# Characterization of Stem-like Circulating Tumor Cells in Pancreatic Cancer

**DOI:** 10.3390/diagnostics10050305

**Published:** 2020-05-15

**Authors:** Lei Zhu, Barbara Hissa, Balázs Győrffy, Johann-Christoph Jann, Cui Yang, Christoph Reissfelder, Sebastian Schölch

**Affiliations:** 1Department of Surgery, Universitätsmedizin Mannheim, Medical Faculty Mannheim, Heidelberg University, 68167 Mannheim, Germany; Lei.Zhu@medma.uni-heidelberg.de (L.Z.); Barbara.Hissa@medma.uni-heidelberg.de (B.H.); Cui.Yang@umm.de (C.Y.); christoph.reissfelder@umm.de (C.R.); 22nd Department of Pediatrics, Semmelweis University, H-1094 Budapest, Hungary; gyorffy.balazs@yahoo.com; 3TTK Cancer Biomarker Research Group, Institute of Enzymology, H-1117 Budapest, Hungary; 4Department of Medicine III, Universitätsmedizin Mannheim, Medical Faculty Mannheim, Heidelberg University, 68167 Mannheim, Germany; Johann-Christoph.Jann@medma.uni-heidelberg.de; 5German Cancer Consortium (DKTK) & German Cancer Research Center (DKFZ), 69120 Heidelberg, Germany

**Keywords:** pancreatic cancer, PDAC, circulating tumor cells, CTC, stem cells, stem-like, stemness, adherens junctions, epithelial-mesenchymal transition, EMT

## Abstract

Pancreatic ductal adenocarcinoma (PDAC) is the fourth most frequent cause of death from cancer. Circulating tumor cells (CTCs) with stem-like characteristics lead to distant metastases and thus contribute to the dismal prognosis of PDAC. Our purpose is to investigate the role of stemness in CTCs derived from a genetically engineered mouse model of PDAC and to further explore the potential molecular mechanisms. The publically available RNA sequencing dataset GSE51372 was analyzed, and CTCs with (CTC-S) or without (CTC-N) stem-like features were discriminated based on a principal component analysis (PCA). Differentially expressed genes, weighted gene co-expression network analysis (WGCNA), and further functional enrichment analyses were performed. The prognostic role of the candidate gene (*CTNNB1*) was assessed in a clinical PDAC patient cohort. Overexpression of the pluripotency marker *Klf4* (Krüppel-like factor 4) in CTC-S cells positively correlates with *Ctnnb1* (β-Catenin) expression, and their interaction presumably happens via protein–protein binding in the nucleus. As a result, the adherens junction pathway is significantly enriched in CTC-S. Furthermore, the overexpression of *Ctnnb1* is a negative prognostic factor for progression-free survival (PFS) and relapse-free survival (RFS) in human PDAC cohort. Overexpression of *Ctnnb1* may thus promote the metastatic capabilities of CTCs with stem-like properties via adherens junctions in murine PDAC.

## 1. Introduction

With 57,000 new cases and 46,000 deaths annually, pancreatic ductal adenocarcinoma (PDAC) is the 9th most frequent malignant disease in males and 8th most prevalent in females and the 4th most frequent cause of death in both genders [1]. At the time of diagnosis, only a fraction of patients are amenable to surgical resection of the tumor. However, even patients who undergo complete surgical resection are at a high risk of either local or systemic recurrence [2].

In order to study the molecular mechanisms of pathogenesis and metastasis of pancreatic cancer, several genetically induced mouse models (GEMMs) have been established in recent decades [3,4,5,6]. The most prominent and best-characterized model is the LSL-*Kras^G12D^*, LSL-*Trp53^R172H^*, *Pdx1*-*Cre* (KPC) mouse model [6].

It is now well established that circulating tumor cells (CTCs) are an integral part of the metastatic cascade in malignant disease [7,8,9,10,11,12,13,14,15,16,17,18]. CTCs are shed from the primary tumor, survive in circulation, and ultimately colonize distant organs where they establish clinically overt metastases [9,16]. However, while thousands or millions of CTCs are shed into the bloodstream over time, the number of metastases is several orders of magnitude lower [19]. Therefore, it can be assumed that only a small fraction of CTCs are actually tumorigenic and thus clinically relevant. In order to achieve the invasive phenotype required to leave the primary tumor bulk and enter circulation, CTCs from epithelial tumors undergo a process called epithelial–mesenchymal transition (EMT). During this process, the cells acquire mesenchymal properties (e.g., migratory capability) while downregulating epithelial traits [20,21]. This process is reverted during mesenchymal-epithelial transition (MET). As both EMT and MET dynamically fluctuate, CTCs in circulation exhibit a high degree of plasticity and represent a heterogeneous population consisting of epithelial, mesenchymal, and intermediary cells [20,22].

In a previous study, CTCs from the GEMM (LSL-*Kras^G12D^*, LSL-*Trp53^flox/flox^*
^or *flox/+*^, *Pdx1*-*Cre*) were isolated and submitted to single-cell RNA sequencing. The raw data of this experiment was released to the Gene Expression Omnibus (GEO) database, and it is available for analysis [23]. In previous studies from our own group, we were able to demonstrate that, apart from immune evasive capabilities, CTCs also possess stem cell properties [13,17]. The main goal of this study was therefore to investigate the role of stem cell properties in murine CTCs of PDAC and to further characterize CTCs with stem cell properties.

## 2. Results

Details regarding the study workflow are depicted in Figure 1.

### 2.1. Pancreatic CTCs Can Be Clustered into Stem-Like and Non-Stem like Categories

After pre-processing, we kept 72 samples for the following study, and 11,931 genes were obtained (Appendix A). In order to classify CTCs into subgroups, the top 3000 genes that presented the highest variability in expression were chosen for the principal component analysis (PCA) [19,24,25]. From these 3000 genes, epithelial (Epcam, Krt7, Krt8, Krt18, and Krt19) [23], mesenchymal (Fn1, S100a4 and Vim) [26], stemness/pluripotency markers (Aldh1a1, Aldh1a2, Cd24a, Cd44, and Klf4) [25,27,28] and the proliferation marker Mki67 [29] were included in subsequent analyses.

All 72 individual CTCs could be divided into three clusters (Appendix A, B). However, this study focuses on whether the CTCs have stem-like features or not, a dichotomous classification; therefore, the CTCs were divided into clusters based on stem-like features. According to the PCA loading plot (Figure 2A), the cluster located in the first quadrant (blue dots) exhibited stemness markers and was defined as CTC-S (CTCs with stem-like features) (Figure 2B). The other two clusters were merged (red dots) and named CTC-N (CTC without stemness features. The correlation heatmap with hierarchical clustering showed a similar result (Figure 2C).

### 2.2. Both Epithelial and Mesenchymal Markers are Expressed on CTC-S

To evaluate each cell within their respective subgroups in more detail, we expanded the marker pool, as not all markers of interest were listed in the 3000 most variable genes. EMT markers (*Snai1, Snai2, Twist1, Zeb1,* and *Zeb2*), epithelial markers (*Cdh1, Egfr, Epcam, Krt7, Krt8, Krt18,* and *Krt19*) [23], mesenchymal markers (*Cdh2, Fn1, Itga5, Sdc1, S100a4,* and *Vim*) [20,23], pancreatic cancer stemness markers (*Abcg2, Aldh1a1, Aldh1a2, Cd24a, Cd44, Cxcr4, Met, Prom1, Klf4, Nanog* and *Sox2*) [30,31], and a proliferation marker (*Mki67*) were included to plot the heatmap (Figure 3A). 

A high degree of expression heterogeneity was observed for most markers. Compared with CTC-N, CTC-S cells demonstrated an increased expression of epithelial markers such as *Egfr*, *Epcam,* and *keratins*. Parallelly, the mesenchymal markers *Fn1* [log_2_(fold change) (FC) = 4.07, false discovery rate (FDR) = 1.42 × 10^−5^] and *Vim* (log_2_FC = 6.37, FDR = 5.89 × 10^−10^) were up-regulated in the CTC-S group, resulting in a phenotype with both epithelial and mesenchymal characteristics. Notably, stemness transcripts showed variable expression, while *Aldh1a2* (log_2_FC = 4.98, FDR = 1.61 × 10^−8^) and *Klf4* (log_2_FC = 9.41, FDR = 1.45 × 10^−21^) were enriched in CTC-S, *Cd44* was significantly down-regulated (log_2_FC = −2.57, FDR = 0.012) (Figure 3A).

### 2.3. The Adherens Junction Pathway is Functionally Enriched in CTC-S

Compared with CTC-N, 1475 genes were significantly up- and 247 significantly down-regulated in the CTC-S group (Appendix A, B).

We next constructed a protein–protein interaction (PPI) network of up-regulated genes by STRING (available online: https://string-db.org/, accessed on 19 August 2019) [32], establishing 1412 nodes and 6695 interactions (Figure not shown). We used the cytoHubba plugin in Cytoscape to narrow this number down to 43 genes, which were subsequently defined as hub genes.

The biological domains of biological process (BP), cellular component (CC), and molecular function (MF) were utilized to describe the biological domain of genes enriched in the CTC-S group. The cellular components’ anchoring junction [Gene Ontology (GO): 0070161, fold enriched = 3.56, FDR = 1.55 × 10^−40^] and adherens junction (GO: 0005912, fold enriched = 3.53, FDR = 9.73 × 10^−39^) were significantly functionally enriched. The enrichment of cell adhesion molecule binding (GO: 0050839, fold enriched = 3.34, FDR = 6.48 × 10^−31^) was observed in the MF domain (Figure 3B).

While GO enrichment analysis focuses on the functions of the genes, the Kyoto Encyclopedia of Genes and Genomes (KEGG) pathway analysis mainly reflects the interaction network and the signaling pathways of the requested genes. In this analysis, several signaling pathways were significantly enriched, including adherens’ junction (KEGG: 04520, fold enriched = 3.42, FDR = 1.47 × 10^−5^) and fluid shear stress and atherosclerosis pathways (KEGG: 05418, fold enriched = 2.61, FDR = 2.05 × 10^−5^) (Figure 3B).

In order to gain deeper insight into the molecular biology of the CTC subsets, we used a weighted gene co-expression network analysis (WGCNA) in this study.

After network construction (Appendix A, B), we identified five gene co-expression modules (blue, brown, green, turquoise, and yellow) with sizes between 67 and 1384 genes in the 72 samples (Figure 4A). Both brown and turquoise modules correlated with stemness (correlation coefficient > 0.7, adjusted *p*-value <0.001), but the results of the brown module were more significant (correlation coefficient = 0.95, adjusted *p*-value = 3.6 × 10^−37^).

We therefore used the brown module in the functional enrichment analysis. In the brown module, 645 genes were functionally enriched in the adherens junction pathway (KEGG: 04520, fold enriched = 3.71, FDR = 3.59 × 10^−3^), which is consistent with the results of the analysis of the differentially expressed genes. In addition, anchoring junction (GO: 0070161, fold enriched = 3.27, FDR = 2.22 × 10^−13^), adherens junction organization (GO: 0034332, fold enriched = 4.29, FDR = 8.11 × 10^−3^) and regulation of adherens junction organization (GO: 1903391, fold enriched = 13.11, FDR = 8.12 × 10^−3^) were all significantly enriched again (Figure 4B).

To identify the pathways with the strongest influence on the CTC-S phenotype, we intersected functional enrichment pathways from the results of differentially expressed genes and WGCNA. This resulted in six pathways, including focal adhesion (KEGG: 04510), adherens junction (KEGG: 04520), ECM-receptor interaction (KEGG: 04512), AGE-RAGE signaling pathway in diabetic complications (KEGG: 04933), proteoglycans in cancer (KEGG: 05205) and fluid shear stress and atherosclerosis (KEGG: 05418) (Figure 5A). Since adherens junction organization (GO: 0034332) was also significantly enriched in the GO: BP aspect of functional enrichment analysis of both differentially expressed genes and WGCNA, we identified the adherens junction as a pathway of interest in CTC-S. In addition, double enriched GO terms, such as the regulation of adherens junction organization (GO: 1903391), cell–cell junction (GO: 0005911), and cell–cell junction assembly (GO: 0007043) provide further evidence that adherens junctions are biologically relevant in the CTC-S group.

In order to identify the key genes in the adherens junction pathway and the entire PPI network., hub genes of up-regulated genes in the PPI, genes enriched in the adherens junction pathway of differentially expressed genes and the WGCNA were intersected. The gene *Ctnnb1* (log_2_FC = 3.24, FDR = 2.40 × 10^−4^) was found significant in all three analyses (Figure 5B). The adherens junction pathway map and gene expression heatmap are plotted in Figure 5C and Appendix A.

### 2.4. Ctnnb1 and Klf4 Positively Correlate

Next, we evaluated the correlation between *Ctnnb1* and stemness/pluripotency markers (*Aldh1a2, Met,* and *Klf4*), which were significantly up-regulated in the CTC-S group (Figure 6). Among these, we found that *Ctnnb1* overexpression most significantly correlates with *Klf4* upregulation (*r* = 0.50, FDR = 4.99 × 10^−5^).

### 2.5. CTNNB1 Is a Negative Prognostic Factor in Human PDAC

As reported above, *Klf4* and *Ctnnb1* were significantly up-regulated in murine pancreatic CTCs with stem-like features. To explore whether *KLF4* and *CTNNB1* are differentially expressed between pancreatic tumor and normal tissues, the visualization web tool Gene Expression Profiling Interactive Analysis 2 (GEPIA 2) (available online: http://gepia2.cancer-pku.cn, accessed on 7 March 2020) was utilized [34]. When evaluating The Cancer Genome Atlas (TCGA) data, both *KLF4* and *CTNNB1* were up-regulated in PDAC compared to the matched normal tissues and the Genotype–Tissue Expression (GTEx) data [35] (Figure 7A,B). 

We further investigated whether candidate genes expression levels correlate with clinical prognosis, including overall survival (OS), progression-free survival (PFS) and relapse-free survival (RFS) (Figure 7C–H) in the TCGA cohort [36]. No significant influence of *KLF4* on survival was observed. In contrast, although *CTNNB1* expression does not influence OS (Hazard Ratio (HR) = 1.20, log-rank *p*-value = 0.373), patients with higher *CTNNB1* expression have significantly worse PFS (HR = 1.67, log-rank *p*-value = 0.009) and RFS (HR = 2.65, log-rank *p*-value = 0.023). 

## 3. Discussion

The aim of this study was to identify and characterize pancreatic CTCs with stem-like characteristics and explore potential underlying mechanisms through integrated bioinformatics analysis.

Numerous molecular markers, including *Cd24, Cd44, Prom1* (*Cd133*), and other genes, have been applied in order to define pancreatic cancer stem cells (CSCs) [30,37,38]. The CTC subgroup identified in this study as CTC-S demonstrated heterogeneous expression of stemness and pluripotency markers in comparison to CTC-N, with some markers showing increased expression (*Aldh1a2* and *Klf4*), some with reduced levels (*Abcg2, Cd44, Cxcr4, Nanog* and *Sox2*) and others with no significant change in expression (*Aldh1a1, Cd24a, Met* and *Prom1*). The mutual exclusivity phenomenon may be due to the heterogeneous nature of stem-like cells and stemness markers [39]. Similar results have been published for breast cancer, where *CD44*^high^ / *CD24*^low^ defines a cluster of tumor cells with stem-like features [40]. This mutual exclusivity is confirmed in the here-presented results.

The invasive capabilities (e.g., migration and invasion) of tumor cells are a prerequisite for successful initiation and completion of metastasis. Accumulating evidence suggests that these features can be acquired through epithelial–mesenchymal transition (EMT), which is considered as a hallmark of tumor cell dissemination [31,41]. The CTC-S subpopulation exhibits a mixed epithelial/mesenchymal phenotype, while the CTC-N group exhibits neither epithelial nor mesenchymal markers. This is consistent with previous studies relating EMT to stemness [42,43]. Furthermore, CTCs with partial EMT phenotype or stem-like features may predict unfavorable survival in cancers independently [44].

Epithelial markers (most prominently *EpCAM*) have been widely used as identification and isolation markers of CTCs, and the prognostic value of *EpCAM*-positive CTCs is evident [15,45,46,47,48], but previous data have pointed towards a relative down-regulation of epithelial markers on CTCs [13]. The here-presented results draw a more complex image of *EpCAM* expression in CTCs—it can be speculated that *EpCAM* expression on CTCs is a continuum from *EpCAM*-negative to highly *EpCAM*-positive cells. While, in this work, *EpCAM* seems to be relatively overexpressed on cells with presumably higher metastatic potential, more research on the role of *EpCAM* on CTCs is clearly required.

In theory, the loss of cell adhesion molecules during EMT should also coincide with the down-regulation of genes associated with adherens junctions. While both *Cdh1* and *Cdh2* (generally considered a mesenchymal marker) are downregulated in the CTC-S subgroup, most of the genes in the adherens junction pathway are up-regulated in comparison to CTC-N, resulting in a mixed epithelial/mesenchymal phenotype of CTC-S. This may represent a transitory state allowing CTC-S to undergo functional adaptation during the metastatic cascade [49]. Additionally, this overexpression of adherens junctions may enable CTC-S to bind to endothelial cells prior to extravasation.

Intercellular adhesion has been demonstrated to be crucial for successful metastasis of epithelial cancers [50]. Among the different types of intercellular adhesions, cell–cell adherens junctions are the most common and contribute to cell polarity, tissue architecture maintenance, cell movement limitation, and proliferation [50]. Strong adherens junctions are mediated by the cadherin–catenin complex [51,52,53]. The here-presented data show that the CTC-S cells express low levels of E-cadherin (*Cdh1*), while the other genes from the adherens junction pathway were up-regulated, such as *Rhoa* and *Cdc42*, both of which are necessary for adherens junction maintenance [54,55]. We hypothesized that CTC-S cells are in the process of forming the adherens junction through actin cytoskeleton remodeling [56]. 

Besides, another critical component of adherens junctions, plakoglobin (also known as γ-catenin), was found to be necessary for the formation of CTC clusters and further contributed to the metastatic cascade [57,58]. In summary, adherens junctions are critical to the formation of CTC clusters, which can form tumor-microemboli, ultimately outgrowing to overt metastases [57]. These cellular aggregates have been detected in 81% of PDAC patients with unfavorable OS and PFS [59]. However, it must be taken into account that all 75 pancreatic CTCs analyzed in this study were isolated individually, which means there are no data derived from CTC clusters in this dataset; the validity of the here-presented data concerning CTC clusters is therefore limited.

Numerous studies have explored the *Wnt* signaling pathway, mainly due to the fact that canonical *Wnt* signaling regulates, stabilizes, and promotes the accumulation of the β-catenin by inhibiting its degradation [60,61]. A few studies have demonstrated that *Wnt*/β-catenin signaling plays a crucial role in the plasticity of stem cells [62,63]. Furthermore, *Wnt* signaling is activated in pancreatic CTCs according to the original study of GSE51372 [64]. Unexpectedly, we found no significant correlation between *Wnt* family members and *Ctnnb1* in the pancreatic CTC-S group. In fact, according to our results, these *Wnt* family members are even expressed at low levels (Appendix A). Given that *Klf4* is overexpressed in the CTC-S group, we believe that *Klf4* inhibits *Wnt* signaling in this context [65]. Besides, the β-catenin has been determined to be necessary for cell adhesion and pluripotency, even without Wnt signaling [66].

*Klf4* (Krüppel-Like Factor 4) is enriched in CTC-S, which positively correlates with *Ctnnb1*. As reported, *Klf4* is a versatile marker that can both suppress [67] or support [68] PDAC. Several studies have investigated *Klf4*, which encodes a protein that belongs to the Krüppel family of transcription factors and plays a vital role in maintaining embryonic or induced pluripotent stem cells as well as in preventing their differentiation [28,69,70]. In a recent study, Zheng et al. [71] found that *Klf4* promotes CTC survival by increasing intracellular reactive oxygen species. Intriguingly, *Klf4* can interact with *Ctnnb1* through transcription regulation or direct protein interaction. As Tiwari et al. [72] reported, *Ctnnb1* is one of the direct repression targets of the transcription factor *Klf4*. On the opposite side, β-catenin could also bind with the promoter of *Klf4* through activating the canonical *Wnt* pathway, as reported [73]. However, we found that the canonical *Wnt* family members are expressed at low levels, as previously mentioned. The evidence suggests that the interaction between *Klf4* and β-catenin might be happening through the formation of a protein complex rather than via the transcriptional regulation in the pancreatic CTC-S group. Even though there are studies claiming that *Klf4* inhibits β-catenin [74,75,76], there are other reports showing that the *Klf4* / β-catenin complex is necessary for the self-renewal capacity of stem/cancer cells [68]. The consensus is that the protein–protein interaction between *Klf4* and β-catenin happens in the nucleus [68,74,75,76]. According to previously published results [76,77], the cytoplasmic β-catenin could migrate to the nucleus freely, and there promote the transcription and translation of β-catenin targeted genes in conjunction with transcription factors. Among the β-catenin binding proteins extracted from epithelial cells nuclei, a 55-kDa protein could be identified as being *KLF4* [76]. Furthermore, the de novo synthesis of both E-cadherin and β-catenin increased at a similar rate in order to reconstruct an adherens junction after the proteolytic disruption of this extracellular interaction in epithelial cancer cells [77].

Clinically, *KLF4* expression in the primary tumor does significantly influence survival. *CTNNB1* is a negative prognostic marker for RFS and PFS. For OS, a trend can be seen, but it fails to reach statistical significance. This supports the hypothesis that the high expression of *Ctnnb1* in CTC-S contributes to the formation of adherens junctions, which in turn may promote the survival of CTC-S in circulation [78]. However, it must be taken into account that the clinical validation was performed using RNA-seq expression data of bulk tumor cells from the primary tumor rather than single-cell RNA sequencing of CTCs, which limits its validity.

It is important to reinforce that this study is an in silico analysis of a single pre-existing dataset from a single pancreatic cancer model, since no other high-throughput sequencing datasets of pancreatic CTCs are publically available. To reduce the bias from limited study resources, we combined the analysis of differentially expressed genes, WGCNA, correlation analysis, and added clinical survival data. The limited sample size, and the absence of prognostic data directly associated with CTC expression data, may partially limit the results. With all that under consideration, more preclinical and clinical studies are necessary to confirm our findings. However, despite its exploratory nature, this study offers new insight into pancreatic CTCs with stem-like characteristics.

## 4. Materials and Methods

### 4.1. Data Collection, Data Quality Check, and Normalization

Single-cell RNA-sequencing data of 75 murine pancreatic CTCs with sufficient quality in dataset GSE51372 were downloaded from the Gene Expression Omnibus (GEO) database (available online: http://www.ncbi.nlm.nih.gov/geo/, accessed on 19 August 2019). Gene expression and clinical information profiles for Pancreas Adenocarcinoma (PAAD) patients were obtained from The Cancer Genome Atlas (TCGA) data portal (available online: https://tcga-data.nci.nih.gov/tcga/, accessed on 19 August 2019) [36].

In the pre-process part, an integrated web tool iDEP 9.0 (available online: http://bioinformatics.sdstate.edu/idep/, accessed on 7 March 2020) was employed to analyze the 75 CTCs sequencing data [79]. Three samples with a sequencing depth of less than two counts per million (CPM) were excluded, genes with less than 0.5 CPM in all the samples were also filtered out, and then transformed the remaining 72 samples with EdgeR: log_2_(CPM + c), pseudo count c = 4 [79]. (Appendix A) Mitochondrial RNA (mtRNA) was not used for QC as mtRNA is not contained in the GSE51372 read counts file.

### 4.2. Principal Component Analysis (PCA) 

The top 3000 (approximately equal to 25%) [80] most variable genes were identified by the iDEP 9.0. Then, we used an open-source web tool named ClustVis (available online: https://biit.cs.ut.ee/clustvis/, accessed on 7 March 2020) to perform the PCA and generate the PCA score plot [81]. Imputation was deemed unnecessary since the data were normalized in the aforementioned part. PCA loading data were downloaded from ClustVis and visualized using imageGP (available online: http://www.ehbio.com/ImageGP/index.php/, accessed on 23 April 2020). Next, the corresponding stem markers (PCA loadings) were used to define the distinct PCA score cluster with stem-like features.

### 4.3. Identification of Differentially Expressed Genes 

After pre-processing, read counts of 11,931 genes were used to identify the differentially expressed genes. Data analysis was performed using the R package DESeq2 [82]. |Fold change (FC) | > 2 and a corrected *p*-value, false discovery rate (FDR) < 0.050 (Benjamini–Hochberg procedure) were set as cutoffs.

### 4.4. Weighted Gene Co-Expression Network Analysis (WGCNA)

We constructed a gene co-expression network of the 3000 most variable genes using the *WGCNA* package [83] in R x64 3.6.1 based on Euclidean distance. Before the topological overlap matrix construction, we estimated the soft threshold β. The details are described in Appendix A. To identify the module correlated with stem-like traits, we used the dynamic tree-cut algorithm (automatic single block method) to cluster dendrogram branches into several modules and assigned them colors. The minimum module size is set at 30, and the modules with larger than 0.9 pairwise correlation were merged. Only the module which significantly positively correlated with stem-like traits was included in the subsequent study.

### 4.5. Definition of Hub Genes

The up-regulated genes were submitted to the STRING 11.0 database [32], and the Protein–Protein Interaction (PPI) network was reconstructed via the Cytoscape software, version 3.5.1 or higher [84]. In addition to text mining, other basic settings were selected, such as co-expression, co-occurrence, databases, experiments, gene fusion, and neighborhood as active interaction sources. Besides that, only interaction pairs that scored >0.9 in the network were selected.

We used the cytoHubba plugin in Cytoscape, which provides 12 topological analysis methods including cluster coefficient, degree, the density of maximum neighborhood component, edge percolated component, maximal clique centrality, maximum neighborhood component and six other centralities (betweenness, bottleneck, closeness, eccentricity, radiality, and stress) for ranking nodes [85]. If the analyzed genes ranked in the top 10% in more than eight of the aforementioned methods, they were defined as hub genes.

### 4.6. Functional Enrichment Analysis 

We used g:Profiler (available online: https://biit.cs.ut.ee/gprofiler/gost, accessed on 23 March 2020) [86] to perform functional enrichment analysis. Several enriched pathways of the Kyoto Encyclopedia of Genes and Genomes (KEGG) [87] and functional interpretations of gene ontology (GO) [88], including biological processes (BP), cellular components (CC) and molecular functions (MF), were identified [28]. The enrichment of fold >2.0 and FDR <0.010 were set as cutoffs.

### 4.7. Kaplan-Meier Survival Plot

Pathologically confirmed PDAC samples and their corresponding RNA-seq data from TCGA were included in this experiment [20]. Pre-processed level 3 data generated using Illumina HiSeq 2000 RNA Sequencing V2 and gene symbols *KLF4* and *CTNNB1* were used. For each tumor sample of the patient, the expression level was determined using a combination of MapSplice and RSEM. The individual sample files were merged in R using the *plyr* package [89]. In the next visualization step, GraphPad Prism 8.2.1 was utilized for the Log-rank (Mantel–Cox) test and to generate the Kaplan–Meier plots.

### 4.8. Statistical Analysis

Correlations of gene expression were evaluated with *cor* function (Pearson method by default) in R with the *PerformanceAnalytics* package. We calculated the Benjamini–Hochberg FDR. Only FDR <0.050 was considered statistically significant.

Additional tools used in this study are listed in Appendix A.

## 5. Conclusions

In summary, we performed an integrated bioinformatic analysis of single-cell RNA sequencing data of pancreatic CTCs derived from the GEMM. We identified two distinct cell populations with (CTC-S) and without (CTC-N) stem cell-like properties. Various markers, including EMT-transcription factors, epithelial, mesenchymal, stemness, pluripotency, and proliferation markers, were used to characterize the CTC-S population. The adherens junction pathway was found to be significantly enriched CTC-S. This pathway may be activated by up-regulated *Ctnnb1* (β-catenin) through its conjunction with transcription factor *Klf4*, thus enabling CTC-S to survive in the bloodstream and promote distant metastasis.

To conclude, this study suggests that pancreatic CTCs with stem-like features might survive in the bloodstream and reach target organs due to evaluated *Ctnnb1* expression and the activation of intracellular adherens junctions pathway.

## Figures and Tables

**Figure 1 diagnostics-10-00305-f001:**
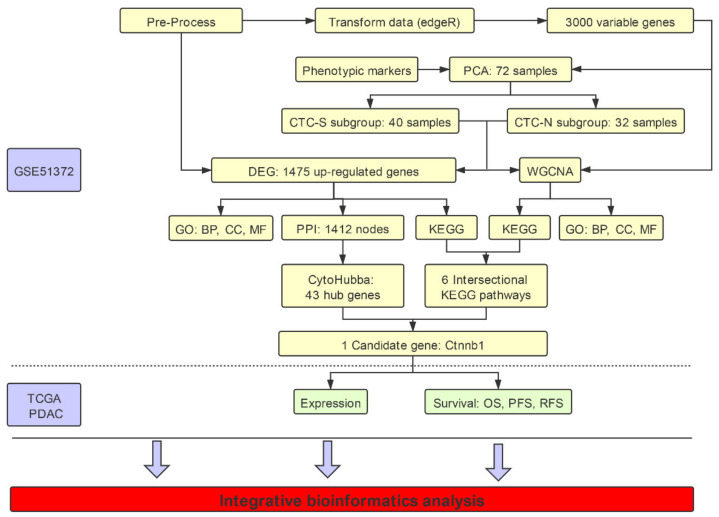
Illustration of the workflow in the integrative bioinformatics analysis. Data set GSE51372 from the Gene Expression Omnibus (GEO) database was used, which contains 75 murine pancreatic CTCs single-cell sequencing datasets. After filtering out three samples with low quality, we further excluded low expression genes, and EdgR transformed the remaining 72 samples. CTCs with stem-like features (CTC-S) and CTC with non-stem-like features (CTC-N) were defined based on the principal component analysis (PCA) of the top 3000 most variable genes. We performed weighted gene co-expression network analysis (WGCNA) on these 3000 genes, while all read counts of 11,931 genes were used to analyze the differentially expressed genes. Further functional enrichment analysis, including the Kyoto Encyclopedia of Genes and Genomes (KEGG) pathways, Gene Ontology (GO) aspects on both differentially expressed genes and WGCNA results was performed. The cytoHubba plugin of Cytoscape 3.5.1 was employed to identify hub genes. Candidate pathways (adherens junction) and genes (*Ctnnb1*) were identified. The clinical value of *KLF4* and *CTNNB1* was validated in the TCGA (The Cancer Genome Atlas) pancreatic adenocarcinoma (PDAC) cohort. The workflow was plotted using ProcessOn (available online: https://www.processon.com/diagrams, accessed on 23 March 2020).

**Figure 2 diagnostics-10-00305-f002:**
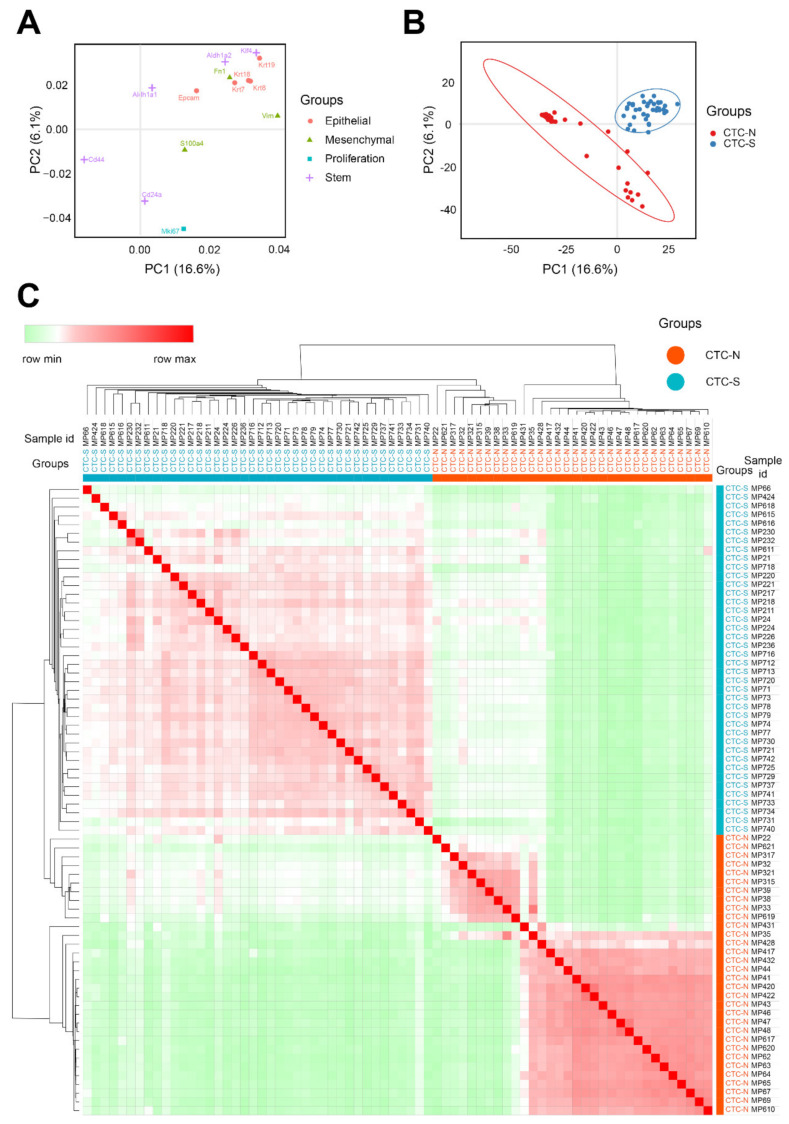
Defining the CTC-S and CTC-N subgroups. (**A**) Principal component analysis (PCA) loading plot of the marker set. The *x* and *y* axes represent the principal components 1 (PC1) = 16.6% variance and PC2 = 6.1% variance, respectively. 3 of the 5 stem markers (Aldh1a1, Aldh1a2, and Klf4) are located in the first quadrant, indicating that stem markers tend to correlate with both PC1 and PC2 positively. The PCA loading data were download from the ClustVis and visualized by the imageGP (available online: http://www.ehbio.com/ImageGP/index.php/Home/Index/index.html,accessed on 23 April 2020). (**B**) PCA scores plot of 72 samples. All samples were divided into three clusters. The cluster located in the first quadrant was defined as CTCs with stem-like features (CTC-S) since they present with stem markers PCA loadings in (A), as the other two clusters were combined and defined as CTC with non-stem-like features (CTC-N). The corresponding ellipses were plotted based on a 95% probability from the same group. (**C**) The correlation heatmap was visualized by MORPHEUS (available online: https://software.broadinstitute.org/morpheus/, accessed on 7 March 2020). We chose the average linkage method to perform the hierarchical clustering. The heatmap demonstrates the distinct subgroups of CTC-S and CTC-N. The colors of the square matrices illustrate the Pearson’s correlation coefficient, with red indicating a strong correlation and green a weak correlation. All samples are listed in the same order in both horizontal and vertical axes.

**Figure 3 diagnostics-10-00305-f003:**
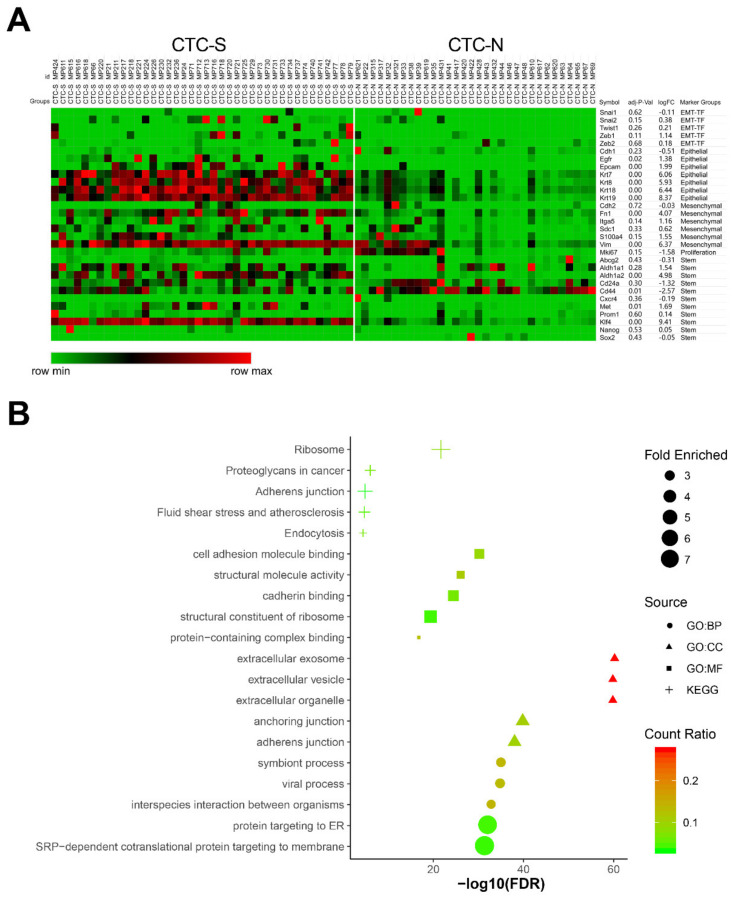
Differentially expressed genes between CTC-S and CTC-N groups. (**A**) MORPHEUS-generated heatmap showing the expression of the marker set in CTC-S and CTC-N groups. The maximum and the minimum values in each row are displayed red and green, respectively. The marker names, groups, log_2_(Fold Change (FC)) values, and false discovery rate (FDR) values were listed on the right side. (**B**) The functional enrichment analysis of up-regulated genes in the CTC-S subgroup was visualized by imageGP (available online: http://www.ehbio.com/ImageGP/index.php/Home/Index/index.html, 23 April 2020), top 5 enriched Kyoto Encyclopedia of Genes and Genomes (KEGG) pathways, Gene Ontology (GO) terms for biologic processes (BP), cell components (CC) and molecular functions (MF), each aspect was shown (fold enriched >2 and FDR <0.010).

**Figure 4 diagnostics-10-00305-f004:**
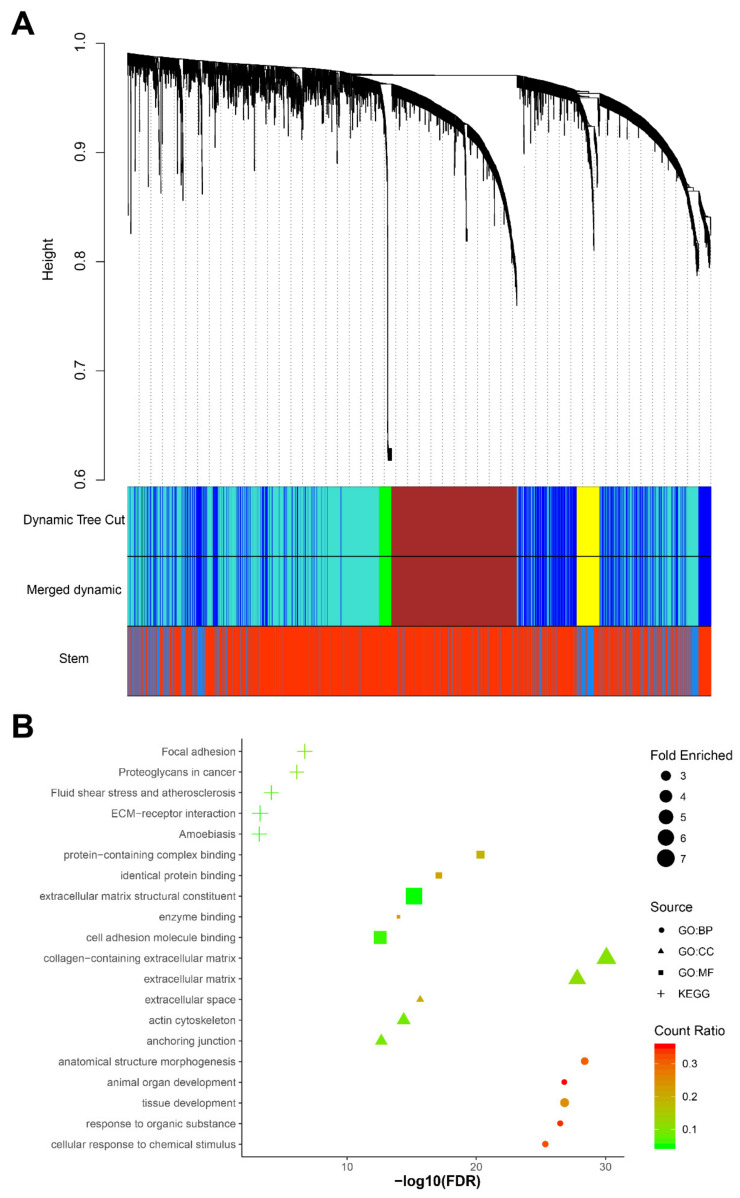
Weighted gene co-expression network analysis (WGCNA) of 72 pancreatic CTCs. (**A**) The cluster dendrogram of 3000 genes, five modules (blue, brown, green, turquoise, and yellow), were identified. The brown module correlates with the stem trait most significantly (*r* = 0.95, adjusted *p*-value = 3.6 × 10^−3^). (**B**) The functional enrichment analysis of genes in the brown module (generated by imageGP, available online: http://www.ehbio.com/ImageGP/index.php/Home/Index/index.html, accessed on 23 April 2020), top 5 enriched Kyoto Encyclopedia of Genes and Genomes (KEGG) pathways, Gene Ontology (GO) terms for biological processes (BP), cell components (CC) and molecular functions (MF) are shown [fold enriched > 2 and false discovery rate (FDR) <0.010].

**Figure 5 diagnostics-10-00305-f005:**
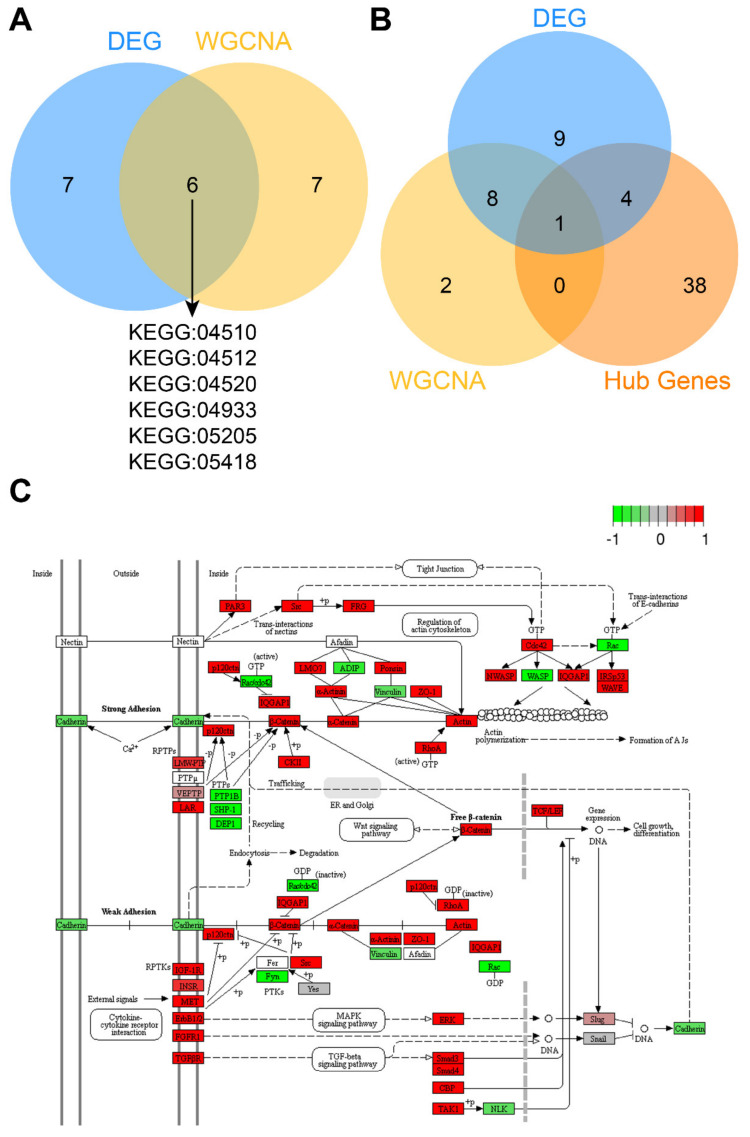
The adherens junctions pathway is significantly enriched in the CTC-S group. (**A**) Venn plot of enriched pathways in differentially expressed genes and weighted gene co-expression network analysis (WGCNA). Six pathways included focal adhesion (KEGG: 04510), ECM-receptor interaction (KEGG: 04512), adherens junction (KEGG: 04520), AGE-RAGE signaling pathway in diabetic complications (KEGG: 04933), proteoglycans in cancer (KEGG: 05205) and fluid shear stress and atherosclerosis (KEGG: 05418) were significantly enriched in both differentially expressed genes and WGCNA. (**B**) The intersection of enriched genes in the adherens junction pathway of differentially expressed genes, enriched genes in the adherens junction pathway of the brown module in WGCNA, and hub genes. *Ctnnb1* is at the intersection of all three lists. (**C**) Gene expression in the adherens junction pathway was plotted by PATHVIEW [33]. Red represents up-regulated genes, while green represents down-regulated genes in the CTC-S group. The Venn plots in this study were generated by imageGP (available online: http://www.ehbio.com/ImageGP/index.php/Home/Index/index.html, accessed on 23 April 2020).

**Figure 6 diagnostics-10-00305-f006:**
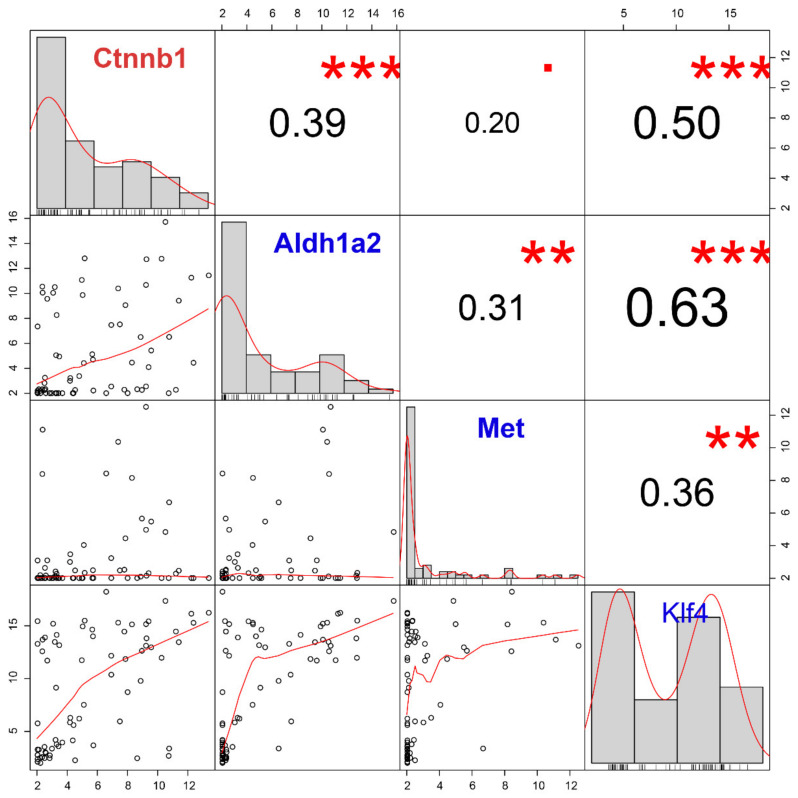
The correlation analysis of the candidate gene (*Ctnnb1*) and stem markers (*Aldh1a2, Met* and *Klf4*) demonstrates the positive correlation between *Klf4* and *Ctnnb1*. Gene expression histograms are represented on the diagonal. The lower left part is the scatter plots and the numbers in the upper right part correspond to the Pearson’s correlation coefficient. *** FDR < 0.001; **, 0.001 < FDR < 0.010;. 0.050 < FDR < 0.100.

**Figure 7 diagnostics-10-00305-f007:**
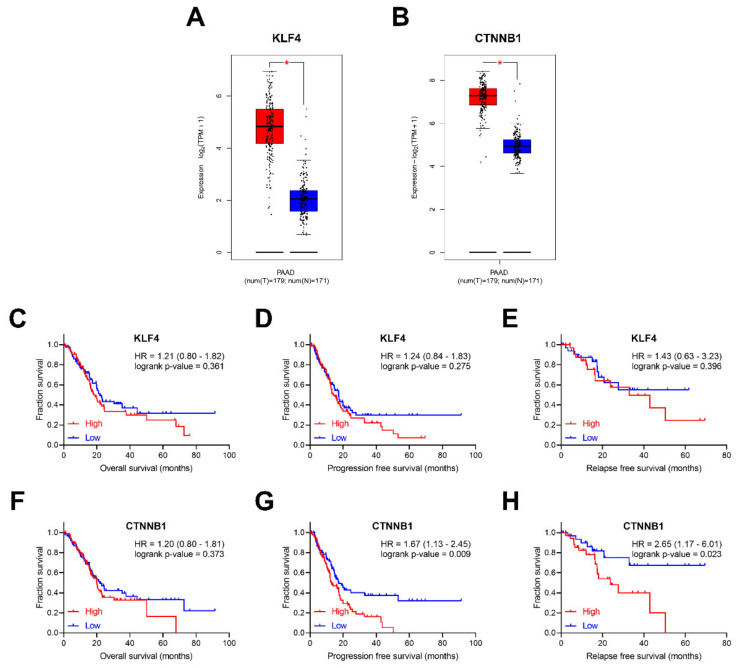
The clinical value of *KLF4* and *CTNNB1*. The box-plots with data overlaid dot plots for *KLF4* (**A**) and *CTNNB1* (**B**), showing expression in TCGA PAAD tumors with correspondent match PAAD normal tissue and GTEx pancreatic tissue. * *p*-value < 0.001. Clinical prognostic value of *KLF4*, overall survival (OS) (**C**), progression-free survival (PFS) (**D**), and relapse-free survival (RFS) (**E**). Clinical prognostic value of *CTNNB1*, OS (**F**), PFS (**G**) and RFS (**H**). Box plots were generated using GEPIA2 (available online: http://gepia2.cancer-pku.cn/#index, accessed on 7 March 2020).

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
