# Peer review of "Characterization of Stem-like Circulating Tumor Cells in Pancreatic Cancer"

_diagnostics, 2020, doi:10.3390/diagnostics10050305_

Round 1
Reviewer 1 Report
Zhu et al. used various bioinformatics tools to characterize stem-like circulating tumor cells (CTC) in pancreatic cancer. The large number of performed analyses make this manuscript difficult to digest. I am also concerned that not enough details are provided to ensure the reproducibility of these analyses.
Specific comments/suggestions/questions:
Make sure to note all parameters and settings used to run the various methods and include justification for all parameter and setting choices. This can be part of the supplement.
If there already are pancreatic cancer stemness markers (lines 114-115), why not use these to define stem-like CTCs in this study?
Top 3000 most variable genes were used to derive PC which were then used to define CTC with and without stem-like characteristics. How was this number (3000) chosen? How does changing it (1000 and 5000) affect PCA results?
Do the PCA clusters correlate with other phenotypes (e.g., age, gender, clinical outcomes, etc.)?
Are the 2 clusters from the PCA plot the same as the 2 clusters in the heatmap (Figure 2)?
Figure 2C: Add legend with color scale. The current legend reads as only positive correlations were found.
Figure 2C: There are clearly 3 clusters. Current assignment into 2 clusters seems artificial and random. Please provide a detailed justification. A dendrogram would also help understanding the cluster structure better.
Figure 3B and 4B: x –axis should be –log10(FDR) not –log(FDR). Make sure to also plot the log10 of FDRs.
Figure 4D is referenced on line 170 but does not exist.
Lines229-231: Given the number of hypotheses tested, these 2 findings should not be over interpreted as they would likely not remain significant if a rigorous multiple testing correction would be performed.
PCA (line 341): Which imputation method was used and how much % missing values were imputed? Please report % missing per genes and per samples.
WGCNA (line 354): How exactly was the Euclidean distance used to identify outliers?
Make sure to reference all software/tools/databases/etc. used in this study. For online tools/databases, make sure to include the date these was accessed.
Reviewer 2 Report
The manuscript entitle "Characterization of stem-like circulating tumor cells in pancreatic cancer" identified two distinct cell populations with (CTC‐S) and without (CTC‐N) stem cell‐like properties, and found overexpression of CTNNB1 is associate with worse progression‐free survival (PFS) and relapse‐free survival (RFS) in PDAC.
1. stem‐like features of CTC‐S and CTC‐N cluster need further justification. Aldh1a1, Aldh1a2 and Klf4 are located in the first quadrant of PCA (F2c) and several stem marker genes also expressed in CTC‐N (Fig3a). It means both CTC‐S and CTC‐N show stem‐like features.
2. If stem‐like feature exist in human PDAC samples, It's more promising to explore CTC‐S signature in PDAC data and how is it related with patient survival. GSE51372 is a small dataset, it't critical to validate stem‐like features in other similar dataset or tumor cohort.
3. Fig7A miss legend
Reviewer 3 Report
This manuscript investigates and characterises the stemness of circulating tumor cells using the publicly available GSE51372 single-cell RNA-seq dataset of pancreatic ductal adenocarcinoma. The authors essentially develop a detailed bioinformatics pipeline to re-analyse the data. Compared to the previous analysis of Ting et al 2014 (that describes some technical details which are left out here), this pipeline is more detailed, looks at different aspects of the data and attempts to derive useful information that was previously unexplored. The figure describing the pipeline is informative and helpful. It describes the basic steps that led to the identification of two candidate genes, CTNNB1 and RHOA. Looking at the data of an independent patient cohort, the authors conclude that CTNNB1 over-expression may promote metastatic capabilities of CTC with stem like properties via the Adherens junctions in murines.
Apart from the fact that the paper needs some reformatting (e.g. end of p.3 and beginning of p.4 and several half-empty pages) I have several concerns regarding the methodology used. I find that the work has potential and for this reason I suggest major revisions:
- Page 3: I would like to see more details on the data QC. At the moment there is only the plot of library sizes in Fig S1. Is this the only criterion used? What about the number of detected genes and the number of reads mapped to mt-RNA genes?
- Page 3: Using WGCNA on the top variable genes is not advisable by the developers of the algorithm: “We do not recommend filtering genes by differential expression. WGCNA is designed to be an unsupervised analysis method that clusters genes based on their expression profiles. Filtering genes by differential expression… so choosing soft thresholding power by scale-free topology fit will fail.” (FAQ of WGCNA package). Please allow me to elaborate on this below.
- Page 3: The authors claim that the PCA (Fig2A) and the heatmap (Fig2C) confirms two distinct clusters. Looking at the data in an unsupervised way, this is clearly not true or, at least, not trivial from the data. The red dots on the PCA span across the whole PC1 range (the largest % of variability), so using the red ellipse to cluster them together is misleading. Possibly, a clustering algorithm such as Affinity Propagation or a hierarchical clustering on the heatmap will reveal 3 groups (as shown in the heatmap the CTC-N cells are split in two groups).
- Page 5: There are several markers listed here most of which make biological sense. However, looking at Fig 3A, not all of them are informative (e.g., the first 6-7 and the last two genes show no differences across the 3 (?) cell clusters). It would be nice to specify which markers are informative for this study and show some evidence of their expression levels on the PCA scatterplot where each dot will have a colour gradient associated to the magnitude of expression (so that we see if certain parts of the PCA are “lit up”).
- Page 7: I do not understand which part of Fig 4A show the existence of two outliers. My guess is that the outliers are shown in Fig S2A. Still, I am not sure how one can label a sample as an outlier using only a dendrogram. Couldn’t these two samples belong to another (very small due to having only 72 high quality single-cells) small cluster? What are the characteristics of these cells other than the library sizes (that according to Fig S1 are quite high) and why are we positive that they are outliers?
- Page 8: I would like to see a more detailed discussion of the “Dynamic tree cut” and “merged dynamic” of Fig 4A. How it was done and what criteria have been used.
- Page 9: The authors found that WGCNA is consistent with the differential expression analysis. In my opinion this is expected since the authors used the top 3,000 variable genes as input. Given that there are (possibly) 3 cell clusters in the data, these 3,000 genes should describe the differences, i.e. differential expression, among the clusters (I don’t see any other obvious source of variability). This might also be the reason that Fig S2B show the dramatic drop in scale free topology at power 9 (maybe it is the failure that WGCNA developers discuss).
- Pages 9-11: although it is clear how the authors identified the Adherens junctions, I am not completely convinced by the result. First, they are based on integrating the results of differential expression with WGCNA but, as described above, the implementation of the latter does not follow the developers’ suggestions. Second, I do not understand the sentences: “We also intersected genes enriched in the Adherens junction pathway of differentially expressed genes, WGCNA and hub genes of up‐regulated genes in the PPI… were found to be up-regulated in all three analyses”. What exactly was intersected? What type of up-regulation refers to WGCNA and PPI? Third, I think that the best way to retrieve such important pathways are by adding Gene Set Enrichment Analysis into the integration strategy (instead of simply the differentially expressed genes).
- Page 11: Fig 6 shows that the most significantly correlated pair is the Klf4-Aldh1a2 with coefficient 0.63. Why do the authors consider the pair with the lower 0.50?
- Page 12: The P-values of Fig 7 are uncorrected for multiple testing (testing for different genes and different types of survival if they are to be used). Regarding the genes, the correction should be done for the whole gene set of the study. In view of this, declaring CTNNB1 as a prognostic markers using the Cox PH model P-value of 0.009 is not well-supported. Which is the gene with the top High vs Low differences in the whole study and where is this gene found in the scRNA-seq data?
- Page 15: Why did the authors use c = 4? My understanding is that c = 1 implies that logCPM (0 + 1) = 0 which is easily interpretable.
- Page 15: Why did the authors normalised the data with EdgeR and then used DESeq2 for differential expression? The model specifications differ but the functionality is the same and the final result is highly reproducible across the two methods. It is also confusing that the authors state “After normalisation, relative expression values of 11934 genes were used to identify the differentially expressed genes”. The models do not use the normalised data but the raw counts. How did the authors use the differential expression model?
- Page 15: I find the PPI criteria too broad. In my experience, the most important ones to obtain a sensible network is experimental validation and interaction pairs scored at least >0.7 (high) or >0.9 (very high). Why did the authors used essentially all types of criteria in this study?
Round 2
Reviewer 1 Report
The authors thoroughly addressed my concerns. I have no further comments or suggestions.
Author Response
Thank you for your revision, which helps us improved the manuscript quality.
Reviewer 2 Report
Comments were well addressed by authors. few minor question:
1. CTNNB1 is a canonical Wnt signaling pathway gene, and The Wnt/β-catenin pathway were reported to play an important role in pancreatic carcinogenesis by regulating cell cycle progression, epithelial-mesenchymal transition (EMT). Could you check if Wnt/β-catenin pathway overall enhanced in CTC‐S ?
2. I'm wondering if TCGA PRAD samples also have CTC‐S and CTC‐N like subpopulation? CTC‐S and CTC‐N feature genes could be used to cluster or score TCGA data.
3. Duplicated Figure2 in revised manuscript
Reviewer 3 Report
Dear Authors,
Thank you for the detailed response to my comments and the extensive reformatting of the paper. I agree with the way you addressed most of my questions. You have done a lot of useful revisions and checks but I find some small problems in the analysis that still need to be addressed.
- There are several duplicated figures in the revised document (due to the way you did the reformatting) and that is confusing. It is often unclear which figures are discussed. Please make sure that in the final document you use the figure that is described in the text.
- The figure legends in the supplementary material are not placed close to the figures which is confusing.
- For points 1 and 2 above, I would appreciate if the authors submitted ‘cleaner’ versions of main and supplementary documents, i.e. without tracking of the changes.
- Please provide a consistent QC analysis in the supplement. At the moment you only have it in your response. Please show the number of reads mapped on the mt-RNA genes (should be less than 30 such genes in a typical RNA-seq experiment based on the gene annotation). Please clarify what you did to get 0 counts in the current figure (did you check the reads of the mt-RNA annotated genes?)
- Please explain in the supplementary what the 'nCount_RNA' plot shows (the one you sent in your response document) and why it is different from the current Supplementary Figure 1A. Which of the two plots was used for the QC? In the present form I am afraid that your QC is quite confusing but, given the fact that I have seen the actual data from the original publication, I tend to believe that the cells you used are of good quality.
- Please show what percentage of your 3,000 top variable genes were found to be differentially expressed in your downstream analysis with DESeq2.
- Figure 2B (the PCA) is also presented as Supplementary Figure 3A with a different number of clusters. This number of clusters provide a better explanation of the data variability (and it has been also used in the original publication). In my humble opinion, the 3-clusters solution is in agreement with the rationale of a single-cell RNA-seq study: find cell sub-populations within a single set of seemingly homogeneous cells (and explain these differences, characterize the cells etc). If the authors choose not to take this argument into account then please explain the reasons to use single-cell RNA-seq on this study instead of bulk RNA-seq and what significant insights they hope to gain by it.
